# Innovative Data-Driven Energy Services and Business Models in the Domestic Building Sector

Juan Aranda [1,*] , Tasos Tsitsanis [2], Giannis Georgopoulos [3] and Jose Manuel Longares [1]

1   Fundación CIRCE, 50018 Zaragoza, Spain
2   Suite5 Data Intelligence Solutions Limited, Limassol 3013, Cyprus
3   Elin VERD SA, 145 61 Kifisia, Greece
*   Correspondence: jaaranda@fcirce.es

**Abstract:** The market of energy services for the residential sector in Europe is very limited at present. Various reasons can be argued such as the high transaction costs in a highly fragmented market and the low energy consumption per dwelling. The rather long payback time for investments render Energy Services Companies' (ESCOs) services financially unattractive for many ESCOs and building residents, thus hindering a large potential of energy savings in a sector that is responsible of almost half of Europe's energy consumption. If the ambitious 2030 and 2050's decarbonisation targets are to be met, the EU's residential sector must be part of the solution. This paper offers insights about novel ESCO business models based on intensive data-driven Artificial Intelligence algorithms and analytics that enable the deployment of smart energy services in the domestic sector under a Pay-for-Performance (P4P) approach. The combination of different sources of energy efficiency services and the optimal participation of domestic consumers in aggregated demand response (DR) schemes open the door to new revenue streams for energy service providers and building residents and reduce the hitherto long payback periods of ESCOs services in the sector. Innovative business models for ESCOs and demand flexibility Aggregators are thoroughly described. Especially customised Performance Measurement and Verification protocols enable fair and transparent P4P ESCO contracts. The new human-centric energy and non-energy services increase the energy consumption awareness of building users and deploy behavioural and automated responses to both environmental and market signals to maximise the economic benefit for both energy service providers and consumers, always respecting data protection rules and the consumers' comfort preferences. The new hybrid business models of P4P energy services make traditional EPC more attractive to energy service providers, with low cost data collection and treatment systems to bring payback periods below 10 years in the residential building sector.

**Keywords:** ESCO business models; aggregator business models; energy services for buildings; energy performance contract; Pay-for-Performance contract; energy efficiency in buildings; demand response; demand side flexibility; residential sector

## 1. Introduction

Despite the large economic energy saving potential in the EU [1], the Energy Service Companies (ESCOs) market is less developed at the domestic building sector compared to other sectors such as the industrial or service building sectors [2,3]. Around 80% of the ESCO market is focused on public buildings, mainly educational and health-care facilities and municipal and regional buildings. The residential building sector is out of the scope of ESCOs to a large extent [4]. Energy Performance Contracting (EPC) providers have been most active in the services and the public building sector, since they are mainly targeting energy contracting offerings to large customers, partly explained by the large transaction costs of energy performance contracts [5].

There are, indeed, specific barriers that make a large-scale application of the conventional ESCO model for residential buildings particularly difficult [6] apart from sector-related barriers (low per-unit consumption, few identifiable homogenous units, lack of the necessary energy intensity to justify investment in today's EPC business models [7,8]). In addition, the decentralised structure of the residential sector hinders the uptake of EPC [9].

However, the increasing penetration of smart solutions for residential dwellings (smart home technologies) and the generation of huge data streams that can facilitate better knowledge of the demand side, the drastic cost reduction for on-site generation and storage, the proliferation of self-consumption models and energy communities, together with the growing de-centralization of the energy system that intensifies the need for introduction of small residential consumers in smart grid management strategies, point the way towards the definition and deployment on innovative energy services that can transform small residential consumers as active energy actors and equal participants in progressively opening energy markets [10,11].

Such favourable conditions are further enhanced by the political commitment at EU [12] and national levels for the empowerment of small consumers to become active elements of the future energy system and an integral part of the integrated EU Energy Market, thus necessitating for new business models and services that can facilitate this transition of the EU energy landscape. Nevertheless, with the drastic reduction in technology costs and the opportunity raised for the creation of significantly high revenue streams through energy markets, it becomes evident that a new era arises for the residential buildings market associated with very attractive payback periods for targeted investments towards energy efficiency, self-consumption optimization, and provision of services to energy grids through demand responses and flexibility [13].

In this sense, an analysis of the framework conditions reveals important factors that can impact the technology deployment in the market in the future:

- Political and legal factors: The concepts above are developed in the framework of the new Energy Efficiency Directive (EED) [14], which establishes a set of binding measures to help the EU reach the 32.5% energy efficiency target by 2030. However, although this directive aims at boosting energy efficiency investments, its transposition, along with other national and local regulations, have posed barriers to ESCOs activities and particularly to the utilisation of the EPC model. These barriers include legal aspects related to the installation of equipment, procurement procedure rules for public authorities, legal issues with tenancy laws, etc. [15]. On the other hand, there are still major barriers in the demand response market in Europe that need to be addressed in many EU Member States [15] due to the low and different pace of the implementation of the internal Union electricity market directives [16]. Despite the latest progress and an increasing regulatory interest [17], the market is still quite fragmented, and this could limit the adoption of some of the energy services in some Member States (for example, the provision of flexibility and demand response services is not regulated yet in many countries) and hinder their overall viability. In addition, the lack of standardised procedures for the measurement and verification of energy savings can lead to disputes between participants due to the impossibility of credibly proving the energy savings, thus limiting the adoption of energy efficiency services [18].
- Economic factors: Despite the untapped energy saving potential in residential buildings [19], there are still economic barriers that impede the adoption of energy efficiency services. One of the main important ones are the high transaction costs for ESCOs compared to the small amount of energy savings (and incomes) achieved per residential household [20], making the economic return lower than in other sectors. While the digital solutions are not very capital-intensive, dispersion of the residential consumers implies that the number of devices to be installed are higher than other sectors (especially for buildings with low smart readiness), which could limit the attractiveness of the services. This is also affected by energy price volatility, since this has a signifi-

cant impact on the feasibility of many energy efficiency measures. In particular, low energy prices make it difficult to guarantee short-term returns on energy efficiency investments. This could be intensified in some Member States, where energy prices are highly subsidised, hence, there is no incentive from consumers to reduce their energy consumption.

- Social factors: Thus far, consumer trust and behaviour towards ESCOs and EPC models has been one of the main limiting factors in the adoption of these services in the residential sector [17]. While ESCOs often work in a context of well-defined user behaviour—the public and tertiary sector—the unpredictability of residential consumer behaviour hampers the implementation of their services due to the risk associated with the volatility of incomes and more difficulties to reach long-term agreements [21]. Hanssen [22] concluded that user behaviour on energy demand is at least as important as building physics. However, general distrust on EPC models and limited confidence in ESCO services are also important barriers for the adoption of energy services in the residential sector. In this framework, the new service bundles aim at overcoming these limitations by developing and enhancing techniques for consumer profiling and savings Performance Measurement and Verification (PMV), thus minimising the risk associated with the volatile behaviour of residential consumers.

- Technological factors: The need for the implementation of digital technology in most of the residential buildings, as well as their fine-tuning to comply with the energy services characteristics, may slow down the adoption of these services in the short term [23]. For old buildings in particular, the potential need for extra data collection equipment would lead to higher CAPEX despite the higher expected energy saving potential.

- Environmental factors: Some authors predict that consumer energy demand will definitely rise in the years coming up to 2050 [24]. The adoption of new energy services will lead to lower energy consumption—both fuel and electricity—at a consumer level, while, at the same time, easing the integration of renewable energies in the grid—thanks to flexibility services—and the implementation of distributed energy generation [25]. Consequently, these services will contribute to reducing GHG and pollutant emissions, hence, there are no significant environmental barriers limiting developments.

The novel services presented in this paper are based on services that use enhanced Performance Measurement and Verification (PMV) protocols to construct continuous consumption and generation baselines based on real-time data (metering and sensing). This PMV is the foundation of the Pay-for-Performance (P4P) principle of the new energy services that goes beyond the current traditional EPC in use by ESCOs for energy retrofitting solutions in buildings, as depicted in Figure 1.

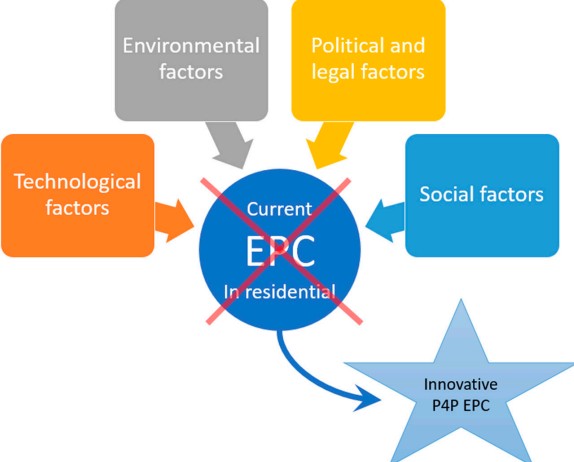

**Figure 1.** Factors hindering the penetration of EPC in the residential building sector.

The paper begins by describing the Pay-for-Performance (P4P) approach applied to the provision of energy services in the residential sector in Section 2. A proposal of innovative energy service bundles is made in Section 3, targeting energy efficiency services, demand response services to the grid, and non-energy services under a P4P perspective. Consequently, the paper describes how the new services can be exploited in two new business models for ESCOs and demand side Aggregators in Section 4. To conclude, a comprehensive development of a tailored PMV methodology is exposed in Section 5, to allow for a fair and accurate verification of the energy and non-energy service performance.

## 2. The P4P Approach

Pay-for-Performance (P4P) programmes have existed for more than 20 years, in different forms, primarily targeting the commercial and industrial building sectors, mainly due to wide smart metering penetration in these sectors [25]. Such programmes have been, in their majority, utility-driven, focusing on building retrofitting performed on behalf of the utility by 3rd parties, which, in turn, receive incentive payments for the actual savings achieved over time and at specified milestones.

On the other hand, historically and to date, there have been just a few programmes oriented towards the residential building sector. Non-availability of fine-grained metering data from residential buildings and loads has been the main reason that has hindered the penetration of P4P into the residential buildings sector. Moreover, available programmes are targeting individual energy efficiency measures that are focusing on achieving energy savings from the replacement of lighting devices and systems, which can be verified and estimated a priori, though missing, to unleash the energy savings potential of residential buildings from a whole building perspective. Another inefficiency of available P4P programmes is that they mostly focus on and remunerate energy savings achieved through specific measures, missing the system perspective and the new role buildings can be assigned with in the smart energy system through their transformation into active flexibility nodes and their introduction as equal participants in flexibility markets.

With the increasing availability of household smart metering, sub-metering data, and IoT data, the residential sector is faced with new opportunities that can be realized with the design and launch of innovative P4P programmes that effectively combine energy efficiency and flexibility triggering measures. Increased data accessibility and granularity can facilitate the measurement and verification (M and V) of achieved savings and flexibility provided in a transparent and objective manner [26], thus enhancing customers' trust over P4P programmes and services. Advancements in data analysis and AI need to be adopted towards the definition of innovative data-driven methods for the dynamic and accurate baselining of energy performance in residential buildings, with a particular view into the objective verification of achieved savings and the remuneration of flexibility services. Moreover, new business models need to emerge that can effectively combine energy efficiency and demand response (flexibility) services and allow both Energy Service Companies (ESCOs) and Aggregators to engage in new business, improve the attractiveness of their traditional offerings and enhance them with hybrid services, de-risking relevant retrofitting investments and ensuring the viability of their business on the basis of standardized hybrid P4P service contracts.

This novel and enhanced P4P approach lies in the heart of the work presented in this paper, and intends to deliver the next generation of Energy Performance Contracting (EPC) on the basis of (a) hybrid innovative energy services properly combining energy efficiency and demand responses; (b) objective, data-driven, and AI-enabled measurement and verification schemes to ensure the objective verification of savings as well as fair and transparent remuneration of flexibility under the principles of Pay for Performance; (c) synergetic business models between Aggregators and ESCOs; and (d) clear and well-specified legal/contractual provisions.

The ultimate target of the P4P approach presented hereinafter is to extract significant value out of the huge energy efficiency and flexibility potential of the (currently overlooked)

residential sector by leveraging on a single point of contact for both EE and DR services, increasing the attractiveness and transparency of relevant investments and services, while preparing the ground for significant uptake and market penetration through the validation of significantly reduced payback times (in comparison with current P4P programmes addressing the residential building sector) achieved with the creation of new revenue streams from residential flexibility transactions in energy markets.

Our P4P approach introduces a variety of service bundles to be provided by ES-COs/Aggregators, under the Energy-as-a-Service (EaaS) model, towards residential consumers in the frame of hybrid energy efficiency and flexibility offerings under the principle of Pay for Performance. These bundles combine (i) building retrofitting and investments for the installation of smart equipment (metering, sensing, actuating), together with extended offerings for the installation of distributed generation (PV) and storage (batteries) units; (ii) energy efficiency services, spanning behavioural transformation and targeted guidance towards energy savings along with more advanced concepts for net metering/self-consumption maximization through smart automation; (iii) flexibility services (with the introduction of storage and electric vehicles as means for enhancing flexibility); and (iv) non-energy services (comfort preservation, indoor air quality, security, well-being, emergency notification services, etc.). Figure 2 shows a summary of the P4P contracts main elements and associated advantages.

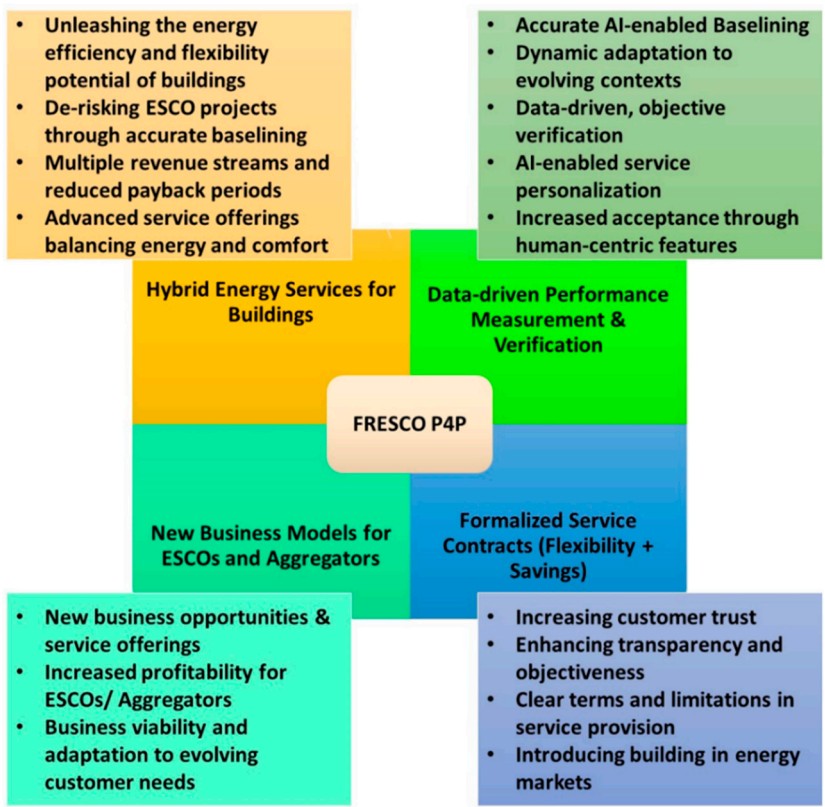

**Figure 2.** The frESCO P4P Approach—Main elements and associated advantages and benefits.

Energy efficiency and flexibility services leverage on the latest advancements in the areas of Big Data Management and AI analytics to safeguard the data-driven nature of the P4P approach introduced, while promoting the provision of truly personalized and human-centric services, ensuring the transparency and objectiveness of measurements and verification through the adoption of a dynamic baselining and normalization approach that relies on accurate long- and short-term forecasting of building energy performance (demand, generation, flexibility). To this end, a standards-based end-to-end interoperability framework has been established, facilitated through an advanced Big Data Management

Platform that effectively addresses and facilitates data collection, management, processing, and analysis of the needs of the variety of services involved in our Pay-for-Performance approach. The Big Data Management Platform enables the interoperable bi-directional communication and data exchange between building amenities and software artefacts utilized for the provision of the P4P services, while preserving data privacy and security through the user-centric definition and assessment of custom rules for data anonymization and accessibility.

The innovative services bundles introduced in our P4P approach are complemented by appropriately drafted business models (for ESCOs and Aggregators) that focus on the establishment of highly profitable business cases for all involved actors by properly extending traditional P4P business offerings. Such new business models aim at allowing ESCOs and Aggregators to individually or jointly provide energy efficiency services combined with flexibility services to the energy system under hybrid service contracts and legal arrangements that ensure attractive payback periods for any investments associated with the services (from IoT device and smart meter/sub-meter subsidies, to generation, storage, and EV incentives).

Settlement of P4P service contracts is performed on the basis of a data-driven Measurement and Verification Methodology that leverages real-time data streams from building amenities to ensure transparent verification of energy savings and flexibility provisions and, respectively, objective remuneration of all involved actors in the realization of the P4P services. Our Measurement and Verification Method is intended to offer fairness, simplicity, accuracy, and replicability in order to foster end users' trust in the remuneration mechanism by properly enhancing existing methodologies and overcoming current barriers (such as the selection of representative days as basis for estimation, the setting of exclusion rules to avoid considering non representative consumption, the definition of adjustment types and windows, and manipulation attempts from the users, etc.). It provides a more objective measurement of the achieved performance for each specific building and an AI-enabled evidence-based definition of relevant energy performance baselines (in the long and short-term), thus ensuring accurate calculation and transparent verification of the provided flexibility and achieved energy savings.

## 3. Innovative Advanced Energy Services

ESCOs and demand side Aggregators are the business stakeholders of the new generation of energy services addressed to the residential sector. ESCOs deliver energy efficiency services while demand flexibility Aggregators valorise aggregated demand response as a service to the grid for balancing and congestion management. In order to maximise the benefits for building users and facility managers the two roles may be played by the same energy service provider utilising a common data collection and managing infrastructure. A proposal of the new data-driven innovative energy services for building users is shown in Table 1.

**Table 1.** Proposal of new data driven ESCO and Aggregator innovative energy services for residential consumers.

| Sensoring and Smart Equipment Retrofitting (RT) | | Comments | Revenue Stream | Expected Output for User |
|---|---|---|---|---|
| RT1 | Smart equipment retrofitting, sensors and meters | Mandatory. Enables all the other services. Include installation, training and coaching | Initial fee, P4P on savings/flexib. | frESCO Big Data platform infrastructure |
| RT2 | Data monitoring and Personalized Informative Billing | Analytics of energy usage and invoicing in a billing period. | Initial fee + Licence fee, P4P on savings/flexib. | Monitoring and data interface |
| RT3 | Smart readiness assessment and Certification | Audit for pre assessment of building efficiency, equipment and smart readiness | service charge | Smart readiness level prior to any new P4P service |

**Table 1.** *Cont.*

| Energy efficiency and self-consumption optimization services (EE) | | Comments | Revenue stream | Expected output for user |
|---|---|---|---|---|
| EE1 | Energy Management for Energy efficiency | Energy efficiency analytics awareness for EE management service. Use of platform data for energy management based on users' comfort choices | P4P on savings | Explicit energy savings |
| EE2 | Personalized Energy Analytics for Energy Behaviour optimization | Implicit EE service. Use of platform data for advice provision (recommendation) and energy mgmt. | P4P on savings | Implicit energy savings |
| EE3 | Holistic self-consumption maximization service | Maximization of Energy self-consumed by Energy management service for prosumers | P4P on savings | energy savings from PV-battery optimization |
| EE4 | Automation and optimal device scheduling | Explicit automated dispatch of efficiency events stemming from EE awareness and price-based scheduling. | P4P on savings | Economic savings |
| Flexibility services (FL) | | Comments | Revenue stream | Expected output for user |
| FL1 | Flexibility analytics services (Awareness and Knowledge of Users' flexibility) | Information towards Flexibility analytics for Flexibility market participation | P4P on flexibility | Flexibility analytics |
| FL2 | Explicit automatic DR services | Implementation of the DR event scheduled. Remuneration for flexibility provisions. | P4P on flexibility | Revenues from ancillary service market or grid operators |
| FL3 | Virtual Power Plant and Optimal Flexibility Activation Scheduling | Schedule of flexibility activation for a future activation. Configuration of VPP | P4P on flexibility | Revenues for flexibility trading in flexibility market |
| Non-energy services (NE) | | Comments | Revenue stream | Expected output for user |
| NE1 | Thermal Comfort services | Comfort preservation and automation at minimum energy costs. Requires smart controls and switches | P4P on service performance | Comfort, automation |
| NE2 | Indoor air quality preservation | Preservation of Indoor air quality by means of air quality sensors. Smart ventilation. | P4P on service performance | Air quality control |
| NE3 | Noise reduction services | Noise sensors. Scheduling of noise devices and appliances at certain periods of time, smart ventilation, others. | P4P on service performance | Noise reduction and control |
| NE4 | Security and surveillance services | Presence sensors, scheduling of lighting at night/absences to create a dissuasive security system | P4P on service performance | Security and surveillance |

### 3.1. New Data-Driven Energy Efficiency Services

The potential of building data usage in energy management systems in buildings is increasingly recognised [27,28]. Energy Efficiency (EE) services focus on obtaining energy savings in different ways from powerful energy analytics to provide the best efficiency strategies for the optimal use of self-produced energy. Two main efficiency strategies are proposed: (a) give recommendations to users for implicit triggering of EE actions, and (b) smart control and scheduling to trigger automatic actions on controllable Distributed Energy Resources (DER). One service, specifically addressed to prosumers, aims at optimising self-consumption from distributed generation assets.

In this regard, the following four energy services are proposed for energy efficiency:

- EE1: Energy Management for Energy efficiency. Energy efficiency analytics awareness for the EE management service. Use of platform data for energy management based on the users' comfort choices.
- EE2: Personalized Energy Analytics for Energy Behaviour optimization. Implicit EE service. Use of platform data for advice provision (recommendation) and energy management.
- EE3: Holistic self-consumption maximization service. Maximization of energy self-consumed by the energy management service for prosumers.
- EE4: Automation and optimal device scheduling. Explicit automated dispatch of efficiency events stemming from EE awareness and price-based scheduling.

These efficiency services, along with additional possible retrofitting measures encompassed in traditional EPC contracts, need a medium to forecast long term demand with a reference period prior to the service deployment, in order to assess holistically the impact of the service deployment.

### 3.2. New Data-Driven Demand Response Services

Aggregated demand responses will play a key role in the future Energy Market [29]. Flexibility services FL are devoted to the provision of demand flexibility from domestic users to be used in grid management in two ways: (a) balancing services to Grid Managers, Distribution System Operators (DSO), Transport System Operators (TSO), and Balance Responsible Parties (BRP); and (b) grid congestion management to alleviate transport and distribution congestion problems at the local and global levels, and the avoidance of costly grid expansion and storage capacity investments made to accommodate an increasing amount of renewable energy sources with high generation uncertainty [30].

In this sense, the following three flexibility services are proposed for this matter:

- FL1: Flexibility analytics services. Information and analytics go towards the awareness and knowledge of the users' flexibility for market participation.
- FL2: Explicit Automatic demand response (DR) services. Implementation of the DR event scheduled. Remuneration for flexibility provisions.
- FL3: Virtual Power Plant and Optimal Flexibility Activation Scheduling. Schedule of flexibility activation for a future activation

### 4. New Business Models for Energy Service Providers in the Residential Sector

The business actors involved in providing these services are ESCOs and Aggregators. These are called the service providers and often a single service provider can adhere to both roles. The main phases in the new energy service value chain are:

- Business opportunity assessment including some pre-analysis and energy audit to assess the efficiency and flexibility potential, and the suitability of the energy services. A personalised service offer may follow.
- Installation of the data collection system in premises. Once the contract has been signed, it is followed by the installation and configuration of sensors, energy meters, data gateways, and smart actuators. Upfront costs in this phase are borne by the service provider.
- Energy service delivery along the contract timeframe. In this phase, a continuous PMV protocol is applied to measure and verify both energy efficiency and eventual flexibility. Revenues are fairly shared between the energy service company and the beneficiary on a regular basis. The sharing rate must ensure the upfront cost payback and the coverage of the operational costs while leaving enough incentives to the end user.
- End of contract. A final settlement is performed to compensate for possible early cancelation expenses and the platform hardware dismantling.

The new cost, revenue, and saving distribution before and during the contract timeframe is shown in Figure 3. Traditional EPC services require a substantial amount of upfront costs with an estimated payback period in the order of 5–15 years, depending on the intervention package. To shorten this period, as many services as possible must be

deployed simultaneously (efficiency, optimisation, flexibility, and non-energy services), since the sunk costs associated with the infrastructure are common, and thus savings or revenues can be maximised.

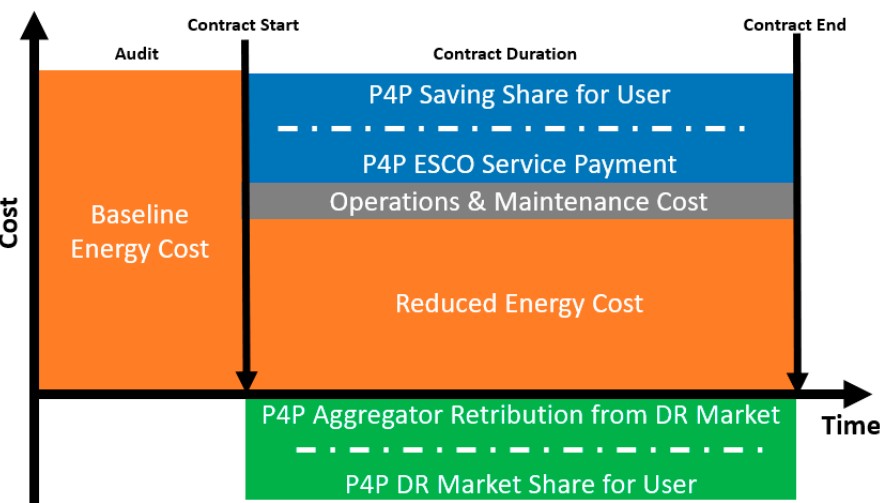

**Figure 3.** Costs, revenues, and savings before and during the contract timeframe.

The maximum exploitation of the infrastructure and relevant optimisation of service deployment greatly impacts the required amount of payback time for the business itself.

It is, hence, extremely important for the service provider to be able to maintain a close to real time monitoring of both the energy consumption and the availability of demand flexibility assets. This feature, combined with direct communication and control are paramount to the effectiveness of the solutions and the maximisation of revenue streams. On the other hand, end-users (usually homeowners) require a control, or an opt-out option, regarding any installed component, in order to maintain their comfort level. This is a conflict-of-interest issue that new business models aim to address by admitting warranties for the capital deployer and the eventual contract duration.

### 4.1. New P4P Business Models for ESCOs

ESCOs primarily provide energy efficiency services. Typically, they operate under an EPC arrangement and are expected to bear the upfront costs of the investment, either directly or indirectly, for example, in combination with a financier. The energy services under the P4P concept provide for:

- Consumer awareness and informative billing as an instrument towards energy savings;
- Increases in the overall value of the facility by smart readiness certification;
- Real time automation and scheduling;
- Retrofitting opportunities;
- Data control, storage and monitoring.

The above services function as components of the revenue streams, which, in turn, are, essentially, savings measured and verified by a robust, fair, and transparent Performance Measurement and Verification (PMV) methodology. The ESCO business model actors and relationships are depicted in Figure 4.

In contrast with a traditional EPC contract, the P4P energy efficiency services provide for centralised information and control to the end-user, the optimization of self-consumption, storage or grid consumption according to market signals, awareness over potential energy savings and informative billing, transparent valorisation of the energy savings, and cash flow monitoring and control.

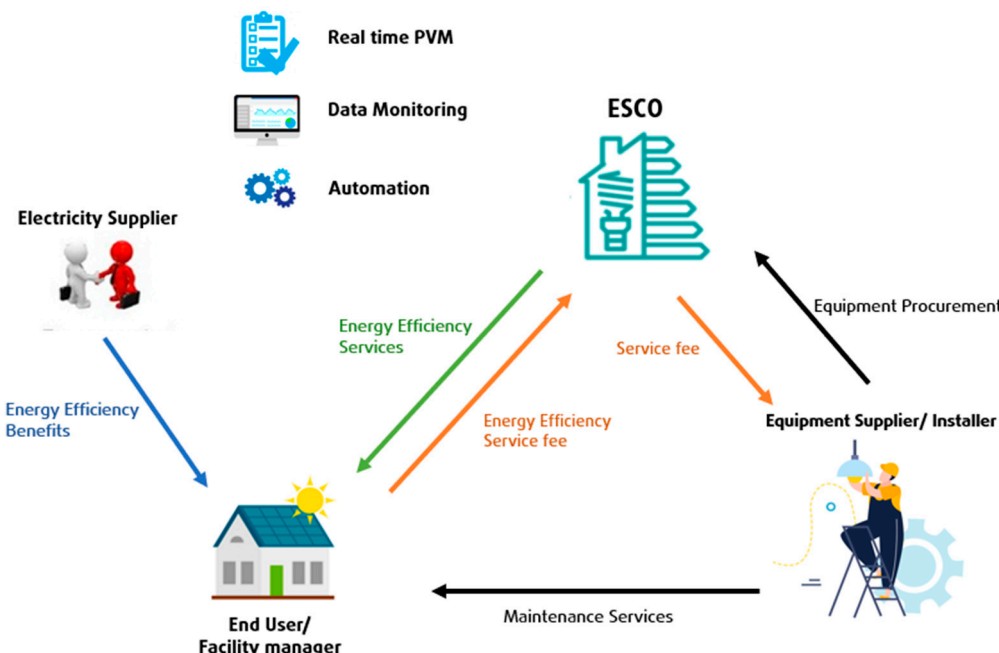

**Figure 4.** ESCO actors and BM.

According to the Business Model Canvas methodology [31], the new ESCO Business Model is summarised in the following points:

Core Values

- Complete services towards the monitoring and controlling, manually or automatically, of local loads, IoT devices as well as generation and storage units.
- Holistic optimization of the energy efficiency of the facility.
- Energy cost reduction.
- Preserving or further enhancing comfort and smart automation.

Services

- Heating, Ventilation, and Air Conditioned (HVAC), and Domestic Hot Water (DHW) control.
- Lighting control.
- Battery control and EV charging.
- Renewable Energy Source (RES) generation and self-consumption optimization.
- Smart Home Automation and scheduling.

Pains Experienced

- Unawareness of market prices and efficiency opportunities.
- Unawareness of real time RES generation and demand matching.
- Manual load micromanagement.

Jobs to complete

- Minimise energy bills.
- Increase energy efficiency and reduce $CO_2$ footprint.
- Automate energy management.

### 4.2. New P4P Business Models for Demand Response Aggregators

Demand side Aggregators on the other hand, are primarily interested in executing flexibility services on demand by grid operators (Explicit DR) or according to price signals from the markets (Implicit DR). Explicit DR implies direct participation in the balancing, capacity, or even wholesale markets. The participants receive direct compensation for providing the energy flexibility required. Implicit DR implies exposure to market or network charges according to the time of use of the electricity. Consumer behaviour is

driven by real-time market signals triggered. In contrast to explicit schemes, the actor is not committed to act but, instead, receive the benefits in the form of a reduced energy bill. Although both demand response schemes are compatible and interesting [32], the business models proposed in this paper refer to explicit schemes to respond to market needs of distributors, network operators, and BRPs.

The Aggregator's objective is to perform peak-load management in active energy markets (flexibility, capacity, energy, etc.) and receive the corresponding market compensation, while maintaining users' comfort levels. Services for Aggregators thus imply hybrid models that effectively combine energy efficiency as well as demand flexibility services that can maximise revenues through the control of small, but significant when aggregated, loads. These services include:

- Predictive flexibility awareness and analytics for buildings.
- Optimal peak demand management.
- Demand side management framework for market participation and VPP configuration.

The flexibility business model actors and relationships are depicted in Figure 5.

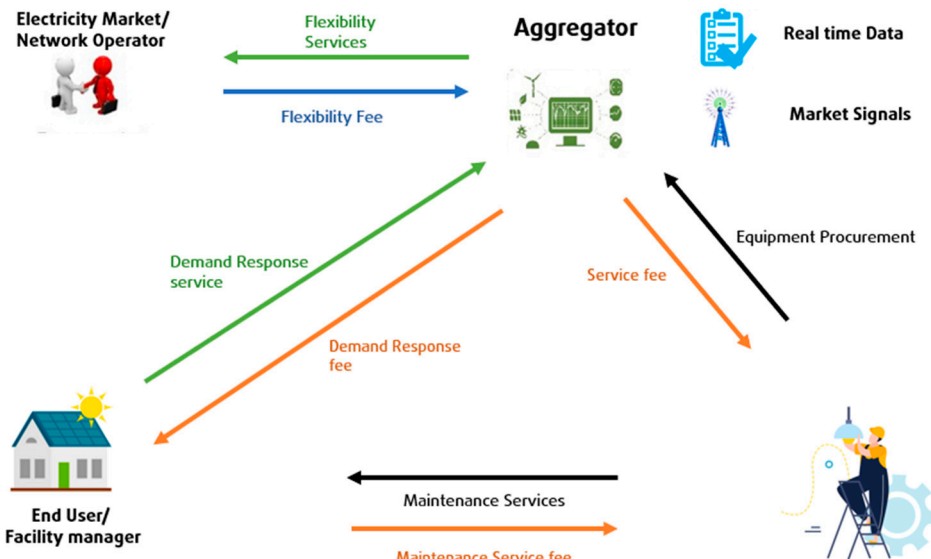

**Figure 5.** Aggregator actors and BM.

The value proposition of the new Aggregator Business Models can be summarised to the following:

Core Values

- Complete services towards monitoring and controlling manually or automatically local loads to participate in the market by offering flexibility services to the grid and earn revenue.
- Utilizing IoT devices as well as generation and storage units towards optimizing the energy value of the facility.
- Improving income while preserving or further enhancing comfort and smart automation.

Services

- HVAC and DHW control.
- Lighting control.
- Battery control and EV charging.
- RES generation and self-consumption optimization.
- Smart Home Automation and scheduling.

Pains Experienced

- Load micromanagement.

- Non-access to flexibility markets without an Aggregator.
- Unawareness of market prices and efficiency opportunities.
- Unawareness of real time RES generation and demand matching.

Jobs to complete

- Gain an overview of residential energy flows.
- Earn extra revenue.
- Increase energy efficiency and reduce $CO_2$ footprint.
- Automate energy management.

In the optimum scenario, building residents contract both types of services simultaneously provided by a service provider that plays both ESCO and demand flexibility Aggregator roles. This hybrid business model adds the savings derived from data-driven efficiency services and the revenues obtained from the participation in open energy flexibility markets to traditional EPC models, thus decreasing the expected payback period of the data platform investment.

Among others, the main factors affecting the economic viability of the new P4P energy services are:

- Upfront costs need to be the lowest possible, leveraging from existing infrastructure and limiting the number of onsite visits for the data capturing infrastructure and gateway.
- The maximum number of services should be contracted simultaneously, thus increasing the revenue sources (efficiency, optimisation, flexibility, and non-energy services) in a hybrid approach.
- Dwellings should have a large electricity demand. Heating, Ventilation, and Air Conditioning should be preferably electric to maximise the potential for savings and demand response [33].
- The service provider customer base should be large enough to share the running costs.
- A substantial part of the benefit sharing should be dedicated to pay for the upfront and running costs incurred by the service provider.

### 4.3. Challenges Faced for the Implementation of the Novel Energy Services

Energy (and non-energy related) services offered under the P4P approach face a series of risks and challenges for their successful market deployment. Technical risks involve cybersecurity and data privacy issues, integration with legacy equipment, the lack of proper maintenance plans, and lack of equipment standardization with different communication protocols and data formats [34].

Economical risks may include the lack of a commercial fit between energy savings services and consumer profiles and habits, meaning that the energy services would not reach their full potential in the long run, impacting the payback period accordingly. From a social point of view, the greatest risk relies on potential lack of interest of consumers to deploy energy services due to the initial cost and performance uncertainty. While periods of high energy prices usually trigger energy efficiency investments, the lack of disposable income due to the high energy expenditure creates significant barriers. Price volatility as such can produce both effects. This risk can be mitigated through the use of a relatively low-cost system of commercially available equipment handling large amounts of data. It enables the provision of a variety of services combining different revenue streams (savings, market remuneration) to fit a vast diversity of buildings with a standard solution.

There are other important challenges the novel business models face:

- Scarcity of valuable data in residential buildings. A high proportion of the current building stock is not prepared to capture and store real time data for AI to deliver valuable services to the building residents. This data includes indoor and outdoor temperatures, humidity, presence, air quality, and metering [35]. The new business models must cater for the necessary data collection and transfer systems such as sensors, meters, and gateways. Smart plugs and actuators for automated services are

also required. The abovementioned equipment is commercially available, standard, and affordable.

- Reliability of AI algorithms. AI analytics and forecasting algorithms are constructed on pretrained models based on the Gradient boosting regression model [36] or neural networks [37]. Model reliability can be affected by many factors such as data granularity and quality. In fact, anomalies inevitably exist in the data. Outlier detection can be triggered by using the maximum likelihood estimation-based Bayesian dynamic linear model [38]. In any case, data filtering and ingestion rules should be defined according to the available data quality to improve the AI algorithm performance. Other commonly used optimization methods of AI performance can be used, such as grid search, random research, and Bayesian optimization methods [39].
- Volatile energy markets. The recent upheaval of the energy markets due to the COVID-19 pandemic and the war in Ukraine has pushed energy prices to unprecedented levels [40]. This situation is boosting innovation in the technical and business aspects of building energy management and driving payback periods down. The high price volatility helps in changing the mindset in an otherwise conservative sector, rather than closing off innovation uptakes.
- The P4P energy service proposal requires low investment needs for data collection and processing equipment and fits in a wide variety of buildings and consumer profiles. Payments based on verified performance are a good guarantee of fair and transparent contractual arrangements for both parties, service providers, and service consumers [41].
- Managerial issues such as the users' fear to intrusion from third parties. These concerns can be derived from the possible misuse of personal data or from the feeling of control loss in favour of AI and automation strategies [42]. To overcome these concerns, automation is limited to device scheduling and short-term explicit demand response events, with prior acceptance by consumers. Data privacy is addressed by implementing the right anonymization and encryption procedures. For this purpose, Blockchain technology is used to encrypt the smart contract information [43] that allows users to digitally sign explicit demand response participation contracts with their demand flexibility aggregator.

Finally, the most important challenge for the deployment of the proposed services is the maturity level of the markets from both an infrastructure point of view as well as a competition perspective between market participants. Smart P4P services require smart meters and smart grid technologies fully deployed by the TSO/DSOs and, at the same time, the existence of smart tariffs and incentives from market participation and engagement. Energy retailers have not fully exploited these approaches, thus the public remains partially unaware of their potential. Local regulation should strive towards the smart grid and dynamic tariff schemes while ensuring consumer protection from critical prices. While this is only a matter of time, a quicker pace towards sustainability may be required.

## 5. PMV Methodologies for Successful P4P Contracts

The P4P contracts require a direct relation between service payment and energy performance. Hence, this performance must be accurately measured and verified. Indeed, P4P energy services are based on a specific PMV Methodology that uses real-time data streams to ensure (a) objective validation and assessment of the feasibility and effectiveness of the new business models, and (b) transparent and fair remuneration of the involved actors for the achievement of energy savings and the provision of flexibility to the grid. The PMV methodology focuses on the establishment of a robust and transparent method based on data streams from local resources and blockchain technology. The new PMV method is based on existing methodologies [34,35], extending the measurement and verification protocols to demand response on the basis of fairness, simplicity, accuracy, and replicability, in order to foster end users' trust in the remuneration mechanism.

Two clear scenarios should be depicted where different methodological approaches are followed: (a) energy efficiency measurement and verification and (b) demand flexibility measurement and verification.

### 5.1. PMV Methodologies for P4P Energy Efficiency Services

The EE PMV aims at measuring energy savings for smart retrofitting and the energy efficiency of new services. Savings are measured holistically at a dwelling or building level in specific periods of medium- and long-time horizons (e.g., billing periods). Savings are directly enjoyed by the end users, and they pay the ESCO a share proportionally to the savings obtained in the period for the service delivery. Savings are achieved from various implicit (behavioural changes) and explicit (automation) strategies, as well as from building and equipment retrofitting. Baselines are built based on historical data comprising a full year. If no historical data are available, baselines are constructed upon pre-selected models and, then, calibrated for a given period using the data flow of energy consumption and the energy consumption driving parameters in the Ex-ante period. Baselines are seasonal to provide a better fit to the actual building energy profile for every season of the year. Baselines are fixed in the reporting period, but they are subject to continuous accuracy checks that may reveal the necessity of non-routinary baseline adjustments calculated periodically at every billing or reporting period. In this sense, EE PMV methodologies are similar to existing protocols in use [36]. Figure 6 shows the ex-ante and ex-post scenarios for energy efficiency measurement and verification.

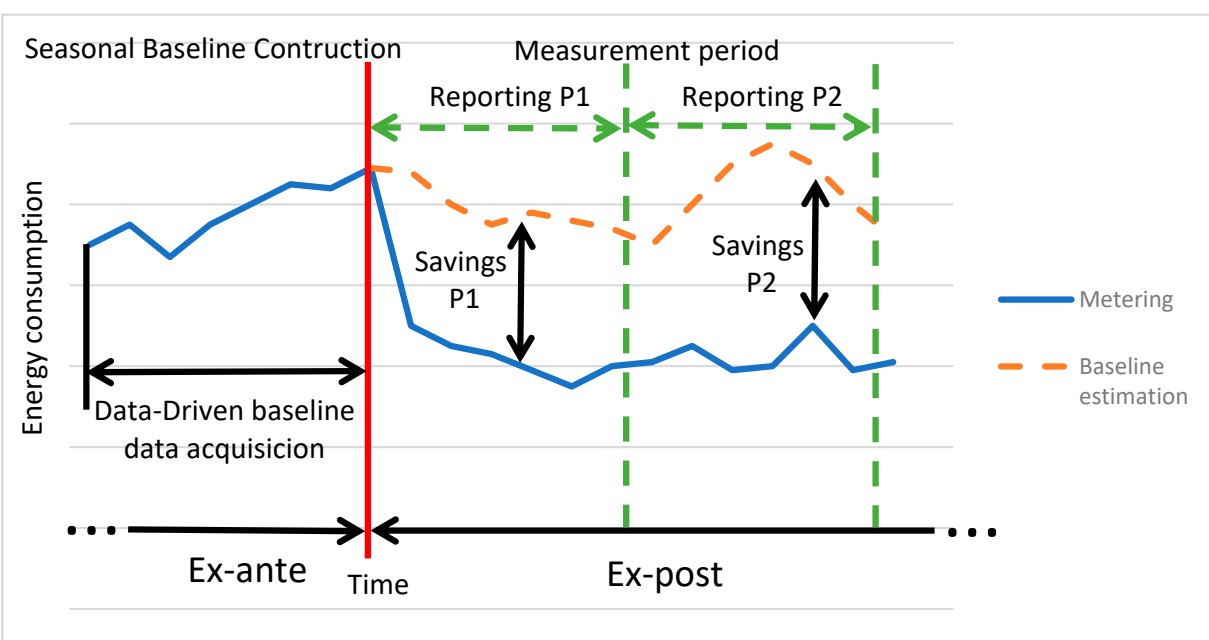

**Figure 6.** Ex-ante and ex-post scenarios for energy efficiency measurement and verification.

### 5.2. PMV Methodologies for P4P Demand Response Services

DR services offer aggregated domestic consumer demand flexibility to grid operators for congestion management and grid balancing services. These services are demanded on short event basis and the performance is measured via the energy shift achieved by the automated operation of available Distributed Energy Resources (DER) by the Aggregator. The Aggregator delivers this demand flexibility to the grid and shares the market remuneration with the flexibility providers or the building users. Flexibility-related events need a short-term forecast so the shortest possible reference period for baseline training is selected while guaranteeing an accurate prediction. Thus, the baseline is load-based and dynamic, as it is recalculated on a continuous basis as new data come into the moving reference or training period. This way, the baseline is updated to any change in external weather

conditions or behavioural changes, avoiding the need of continuous manual adjustments, fitting to the latest user energy profile. Finally, the baseline is calculated using the values of the independent variables just prior to the event, thus reflecting the actual conditions that are the closest to the event. Figure 7 shows the Ex-ante and ex-post scenarios for energy flexibility measurement and verification.

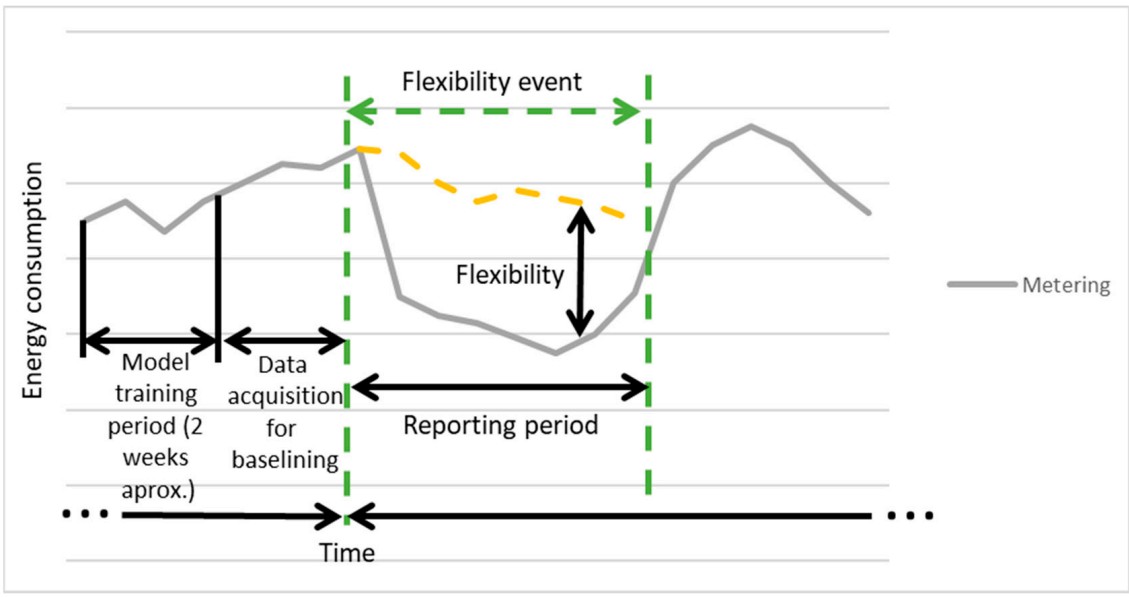

**Figure 7.** Ex-ante and ex-post scenarios for energy flexibility measurement and verification.

### 5.3. Performance Assessment of Non-Energy Services

Non-energy services are an additional value added of data-driven services that use the available user and building data to deliver optional benefits to the end users such as comfort, noise control, air quality, or others, all under P4P contracts. The performance of these optional services is not based on energy measurements but on compliance with the contractual service levels. Measurements of the involved service parameters are compared to the target values to derive service payments from the degree of compliance with the set targets.

### 6. Conclusions

The new energy services described in this paper constitute an innovative portfolio expansion for ESCOs in the residential sector, where these companies hardly have any presence today. Among other reasons, the low EPC penetration in the residential sector is explained by the low benefits derived from a scarce per capita consumption rate and the high transaction costs involved due to the high energy usage fragmentation. These factors and the difficulty to aggregate significant energy amounts to justify an investment in an energy management system has kept ESCOs away from a sector that represents 40% of the EU's final energy consumption.

Today's digital technology can drive energy service costs down and make them available to many domestic consumers. The main contribution of this research is to show how ESCOs can extend the conventional concept of EPC models, traditionally suitable for retrofitting services, to a new set of attractive standard data-driven services powered with AI algorithms. These services can be delivered on a continuous basis in an unsupervised manner, and are extensive simultaneously to a large number of building residents, thus reducing the per-capita operation costs. These offerings are now expanded by novel services to the grid in the form of demand flexibility services for congestion management, grid balance, and ancillary services triggered by network operators in local energy markets. In addition, non-energy services can also be designed to suit consumers needs such as

automation and comfort, air quality, noise control, or safety and surveillance, among others, as value-adding alternatives, delivered with the same infrastructure.

The main problem with EPC services in the current domestic sector is the difficulty of establishing the right performance and verification procedures to ensure a fair remuneration for the services and adapt the verification methodology to non-monitored and changeable independent variables such as outdoor and indoor weather conditions, PV generation, or user comfort preferences. In this sense, the new P4P energy service generation solves this problem by creating a Pay-for-Performance model, where performance is continuously being monitored by means of powerful and accurate near real-time forecast algorithms fed by the digital platform data captured from metering and system behaviour parameters. These P4P models enable a fair distribution of energy savings and flexibility market remuneration that shares proportionally the benefits among all market actors, the service providers on one side (ESCOs and/or Aggregators) and the consumers/prosumers on the other side.

The new set of P4P hybrid energy services combine the traditional EPC retrofitting services with the savings obtained from several implicit and explicit energy efficiency strategies and with the potential remuneration of demand response from flexibility markets triggered by network managers. The combination of these revenue sources derived from the use of the same digital platform can reduce the payback times greatly, below 10 years, thus making them more attractive to residential building managers and residents.

These energy service sets can be provided by either ESCOs or Aggregators, but the ideal scenario for service providers would be to play both roles along with possible non-energy services enabled by the digital platform.

**Author Contributions:** Conceptualization, T.T.; Formal analysis, J.A. and G.G.; Investigation, J.M.L.; Methodology, T.T.; Writing—original draft, J.A.; Writing—review and editing, J.M.L. All authors have read and agreed to the published version of the manuscript.

**Funding:** This contribution has been developed in the framework of the frESCO project 'New business models for innovative energy services bundles for residential consumers', funded by the European Commission under the H2020 Innovation Framework Programme, project number 893857.

**Institutional Review Board Statement:** Not applicable.

**Informed Consent Statement:** Not applicable.

**Data Availability Statement:** The data presented in this study are openly available in www.fresco-project.eu.

**Conflicts of Interest:** The authors declare no conflict of interest.

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
