# Peer review of "Innovative Data-Driven Energy Services and Business Models in the Domestic Building Sector"

_sustainability, doi:10.3390/su15043742_

Round 1
Reviewer 1 Report
This manuscript offers insights about novel ESCO business models based on intensive data-driven Artificial Intelligence algorithms and analytics that enable the deployment of smart energy services in the domestic sector under a P4P approach. The research methods are reasonable, and the results are interesting.
However, some revisions should be made before this work can be accepted for publication.
1. Abstract should be rewritten by addressing actual research content and main findings.
2. The introduction section needs to be expanded in order to provide a useful background for this study. The literature referenced in the background to ESCOs and EPCs is very old. Are the views expressed in the paper in line with current and future developments? Please add references from the last five years to support this.
3. To highlight that your work is an effective work in ESCO business models and to enhance the literature review plus compare your research with other researches in related fields
4. The main contribution should be clarified.
5. Figure 1 should be provided with the appropriate explanation and cited.
6. It is advisable to redraw Figure 2. Please explain why cost is the vertical axis. If there is no functional relationship, suggest changing to a table.
7. In Lines 355, 392, 489, etc, check for punctuation problems.
8. Please explain the meaning of HVAC and DHW in line 396 and RES in line 399.
9. Redraw Figure 5-6.
10. Sections should be added on the specific applications of the proposed data-driven artificial intelligence algorithms and other digital technologies. Otherwise, the abstract and conclusions should be revised.
11. Managerial insights should be suggested by addressing some new meaning full findings.
12. The manuscript should be prepared according to journal guidelines, and should be really carefully edited in References.
Author Response
Dear Sir / Madam
On behalf of the authors, I would like to thank you for your kind comments and remarks to our research. We have rewritten the document addressing the shortcomings identified in your review. We hope that we have satisfactorily responded to the issues raised in the new version of the manuscript and remain at your disposal for any further question or comment from your side.
Please find below our answers to your questions.
- Abstract should be rewritten by addressing actual research content and main findings.
Modifications have been made in the abstract to clarify the main research findings.
- The introduction section needs to be expanded in order to provide a useful background for this study. The literature referenced in the background to ESCOs and EPCs is very old. Are the views expressed in the paper in line with current and future developments? Please add references from the last five years to support this.
No significant changes have affected the ESCO business in the residential sector in the last years. We agree with your comment and new more recent references have been quoted as per your suggestion.
- To highlight that your work is an effective work in ESCO business models and to enhance the literature review plus compare your research with other researches in related fields.
New references have been analysed and quoted to better refer our work with previous similar research.
- The main contribution should be clarified.
The main contribution of the paper is the proposal of innovative business models that leverage hybrid data – driven energy services to boost a new generation of EPC under a P4P approach to be more attractive in the residential building sector. The idea is further enhanced in the Conclusion chapter.
- Figure 1 should be provided with the appropriate explanation and cited.
Figure 1 has been reformulated to summarise the P4P aspects and advantages described in the text. This figure is created by the frESCO project consortium referenced and acknowledged.
- It is advisable to redraw Figure 2. Please explain why cost is the vertical axis. If there is no functional relationship, suggest changing to a table.
Figure 2, now Figure 3 in the new version has been changed to better explain the cost, revenues and savings before and after the contract timeframe. The initial cost of energy, baseline cost, reflects the energy needs of the household before the contract deployment. During the contract the energy cost is reduced and the difference with respect to the baseline costs are used to pay the system operation and maintenance costs and the efficiency service provision under a P4P approach. The rest of the savings estimated with the same approach is enjoyed by the consumer. Additionally, there is a remuneration from the DR market participation (represented as negative costs), that is shared between the DR aggregator and the end user under the P4P principle.
- In Lines 355, 392, 489, etc, check for punctuation problems.
Thank you for the corrections. Done
- Please explain the meaning of HVAC and DHW in line 396 and RES in line 399.
HVAC stands for Heating, Ventilating and Air Conditioned. DHW stans for Domestic Hot Water. Done.
- Redraw Figure 5-6.
Figures 1, 2 5 and 6 have been redone for clarity.
- Sections should be added on the specific applications of the proposed data-driven artificial intelligence algorithms and other digital technologies. Otherwise, the abstract and conclusions should be revised.
New insights about the proposed data-driven artificial intelligence algorithms are provided in the new section 4.3, although it is not in the scope of the paper to go in depth into the AI technology used within the frESCO project developments.
Following your recommendations, the abstract and conclusions have also been revised.
- Managerial insights should be suggested by addressing some new meaningful findings.
We identify as managerial issues and insights the users’ fear to fear to intrusion from third parties. This includes data privacy issues and loss of equipment control driven by AI. This comment is addressed at section 4.3, in the discussion of the “Managerial issues such as the users’ fear to intrusion from third parties ”
- The manuscript should be prepared according to journal guidelines, and should be really carefully edited in References.
All references have been re-edited and expanded, following the journal guidelines provided in the “Guide for Authors”.
Reviewer 2 Report
This paper presented a new ESCO model based on an AI algorithm to benefit energy service providers and consumers. The subject of this paper is interesting, and this paper is well-written. It can be accepted after addressing the following concerns:
1. In fact, anomalies inevitably exist in the data, which can significantly affect the reliability of data-driven models. The authors are suggested to add a brief review about anomaly/outlier detection, and references like Anomaly detection of structural health monitoring data using the maximum likelihood estimation-based Bayesian dynamic linear model should be added.
2. Now that the authors use the AI algorithm to develop the model, what kinds of data do they use? The inputs and outputs are also not clear.
3. The introduction of AI is too brief. For AI algorithms, hyperparameter optimization directly influences their performance. The commonly used optimization method, such as grid search, random research, and Bayesian optimization should be briefly reviewed. The related details can refer to the Probabilistic framework with Bayesian optimization for predicting typhoon-induced dynamic responses of a long-span bridge. This reference should be also added.
Author Response
Dear Sir / Madam
On behalf of the authors, I would like to thank you for your kind comments and remarks to our research. We have rewritten the document addressing the shortcomings identified in your review. We hope that we have satisfactorily responded to the issues raised in the new version of the manuscript and remain at your disposal for any further question or comment from your side.
Please find below our answers to your questions.
- In fact, anomalies inevitably exist in the data, which can significantly affect the reliability of data-driven models. The authors are suggested to add a brief review about anomaly/outlier detection, and references like Anomaly detection of structural health monitoring data using the maximum likelihood estimation-based Bayesian dynamic linear model should be added.
Thanks for the useful suggestion. The reference is added in the new section 4.3 (Ref 38) at the discussion of the “Reliability of AI algorithms” issue.
- Now that the authors use the AI algorithm to develop the model, what kinds of data do they use? The inputs and outputs are also not clear.
A brief summary of the data inputs and outputs is given in the new section 4.3 at the discussion of the “Scarcity of valuable data in residential buildings” issue.
- The introduction of AI is too brief. For AI algorithms, hyperparameter optimization directly influences their performance. The commonly used optimization method, such as grid search, random research, and Bayesian optimization should be briefly reviewed. The related details can refer to the Probabilistic framework with Bayesian optimization for predicting typhoon-induced dynamic responses of a long-span bridge. This reference should be also added.
Thanks for the useful suggestion. The scope of the paper does not extend to the Artificial Intelligence details and rather focuses on the managerial and business aspects of the new generation of services. However, the reference is added in the new section 4.3 (Ref 39).
Reviewer 3 Report
The authors presented new data-driven energy services and business models based on P4P approach. The ideas were discussed clearly and easy to understand. I recommend the paper to be published with minor changes. I have some comments:
1. The title includes both “new” and “innovative” seems redundant.
2. The introduction is rather long though the bold subtitles did help with reading. The ideas are clear but a figure or flowchart would boost the core concepts.
3. Figure 1 is concise and summarized the aspects of the proposed P4P approach. It would be even better to also emphasize the advantages (which were mentioned under various sub points and conclusions) compared to traditional or current methods. The figure doesn’t seem to include much information discussed in the text. Please remove red line under (Flexibility + Savings).
4. Is “3. Innovative Advanced Energy Services relationship” a parallel point to the previous point? Or is it under the P4P approach as well? “4. New Business Models for Energy Service Providers in the residential sector” seems to be under the same P4P approach summary in 2. Should it (or both 3 and 4) be a sub-session under 2 to avoid confusions?
5. The texts in Figure 2, Figure 5, and 6 are blurry on a bigger screen.
6. Would love to see some comments and discussions about how the proposed approach would fit in a constantly changing climate.
7 7. Are there any challenges in implementing such approaches? How would it be avoided or minimized?
Author Response
Dear Sir / Madam
On behalf of the authors, I would like to thank you for your kind comments and remarks to our research. We have rewritten the document addressing the shortcomings identified in your review. We hope that we have satisfactorily responded to the issues raised in the new version of the manuscript and remain at your disposal for any further question or comment from your side.
Please find below our answers to your questions.
- The title includes both “new” and “innovative” seems redundant.
Thanks for the suggestion. Although “new” and “innovative” do not necessarily mean the same, we agree the comment and have deleted the first adjective “new” from the title. Done
- The introduction is rather long though the bold subtitles did help with reading. The ideas are clear but a figure or flowchart would boost the core concepts.
The introduction chapter has been cut down and a new flowchart has been added reflecting the core concepts in Figure 1.
- Figure 1 is concise and summarized the aspects of the proposed P4P approach. It would be even better to also emphasize the advantages (which were mentioned under various sub points and conclusions) compared to traditional or current methods. The figure doesn’t seem to include much information discussed in the text. Please remove red line under (Flexibility + Savings).
Figure 1 has been reformulated to summarise the key aspects of the proposed P4P approach with special focus on the advantages of this approach compared to traditional EPCs. In the revised version this is now Figure 2.
- Is “3. Innovative Advanced Energy Services relationship” a parallel point to the previous point? Or is it under the P4P approach as well?
Point 3, Innovative Advanced Energy Service, proposes a new set of energy service provided under the P4P approach described in point 2.
“4. New Business Models for Energy Service Providers in the residential sector” seems to be under the same P4P approach summary in 2. Should it (or both 3 and 4) be a sub-session under 2 to avoid confusions?
The document starts by describing the P4P methodology in chapter 2, under which a new set of innovative services are proposed in chapter 3. Chapter 4 goes one step ahead describing the proposed business models to offer the services described in chapter 3.
The new Business models cluster the different innovative services offered by ESCOs and DR Aggregators where the retributions and incomes for the service providers are verified and calculated with a P4P methodology.
- The texts in Figure 2, Figure 5, and 6 are blurry on a bigger screen
Figure 2, now Figure 3 in the new version has been changed to better explain the cost, revenues and savings before and after the contract timeframe. The chart quality has been improved for the three charts.
- Would love to see some comments and discussions about how the proposed approach would fit in a constantly changing climate.
See response below.
- Are there any challenges in implementing such approaches? How would it be avoided or minimized?
A new section 4.3 “Challenges faced for the implementation of the novel energy services” has been added with hints about how the barriers are being tackled, among which the everchanging energy environment (point 6 in your list).
Round 2
Reviewer 1 Report
This revised paper can be published in Sustainability.
Reviewer 2 Report
The reviewer's comments have been addressed.